biochemistry/molecular biology

parchment, medieval, proteomics, birth girdle, stains

**Author for correspondence:**
Sarah Fiddyment
e-mail: sarah.fiddyment@palaeome.org

# Girding the loins? Direct evidence of the use of a medieval English parchment birthing girdle from biomolecular analysis

Sarah Fiddyment[1], Natalie J. Goodison[2,3],
Elma Brenner[4], Stefania Signorello[4], Kierri Price[4,5]
and Matthew J. Collins[1,6]

[1]The McDonald Institute, Department of Archaeology, University of Cambridge, Cambridge, UK
[2]The Institute for Advanced Studies in the Humanities, University of Edinburgh, Edinburgh, UK
[3]Department of English Studies, Durham University, Hallgarth House, 77 Hallgarth Street, Durham DH1 1AY, UK
[4]Wellcome Collection, 183 Euston Road, Bloomsbury, London NW1 2BE, UK
[5]Department of English, Theatre and Creative Writing, Birkbeck, University of London, Malet Street, Bloomsbury, London WC1E 7HX, UK
[6]Evogenomics, The Globe Institute Department of Health Sciences, University of Copenhagen, Sølvgade 83, København K, Denmark

SF, 0000-0002-8991-2318; MJC, 0000-0003-4226-5501

In this paper, we describe palaeoproteomic evidence obtained from a stained medieval birth girdle using a previously developed dry non-invasive sampling technique. The parchment birth girdle studied (Wellcome Collection Western MS. 632) was made in England in the late fifteenth century and was thought to be used by pregnant women while giving birth. We were able to extract both human and non-human peptides from the manuscript, including evidence for the use of honey, cereals, ovicaprine milk and legumes. In addition, a large number of human peptides were detected on the birth roll, many of which are found in cervico-vaginal fluid. This suggests that the birth roll was actively used during childbirth. This study is, to our knowledge, the first to extract and analyse non-collagenous peptides from a birth girdle using this sampling method and demonstrates the potential of this type of analysis for stained manuscripts, providing direct biomolecular evidence for active use.

# 1. Introduction

## 1.1. History of the birth roll

Childbearing can be a dangerous time for both mother and child even today [1–3], but in medieval and early modern Europe the risks of childbearing were extremely perilous. While only 9 out of 100 000 women died from childbirth in England in 2013 [4], in early medieval England, childbirth was thought to be the main cause of death for women. In eleventh-century Norwich, skeletal material from a poverty-stricken area reveals the infant mortality rate was over 60% and the average age of death for women was 33 [5]. Even though the quality of life improved over the course of the Middle Ages, particularly for elite women [6], the neonatal mortality rate for mother and child was reckoned to be somewhere between 30% and 60% [7]. As Rawcliffe notes, even the healthiest of women in childbirth had good reason to fear protracted confinement, permanent injury, if not death [5]. The high death-toll for women reflects complications caused by childbearing, including postpartum infection, eclampsia or retained placenta. In addition to this, women and children-in-utero of fifteenth and sixteenth-century Britain would have borne a heightened infectious disease burden, particularly at risk to plague, which continued to outbreak in epidemics in England between 1348 and 1665 [8]. Standards of living, complications in pregnancy and childbirth, and increased risk from disease clearly indicate the act of giving birth was fraught with danger. In fact, even the Book of Common Prayer (1549) stresses this peril for women, praising God for safely delivering women from the 'greate daunger of childebirth' (great danger of childbirth) [9].

Pre-reformation devotion in England indeed encompassed many feminine appeals for safe delivery, which included invoking several female saints, including St Mary (the mother of Jesus), St Anne (the mother of St Mary), St Susanna (a woman falsely accused of adultery) and St Margaret (who was once swallowed by a demonic dragon and burst forth—a sign taken in the Middle Ages of release from the prison of pregnancy) [5]. The anxieties of childbirth extended beyond the hour of birth in the birthing room. Green demonstrates that specific masses were dedicated to St Susanna for the safe delivery of children, which further emphasizes that fears surrounding childbirth permeated rituals and shaped daily routines [10]. It is also worth noting that St Susanna was invoked in other obstetrical texts (British Library, MS Sloane 249) and appears in a charm on another roll held by Wellcome Collection (MS 410, not identified by researchers as a birth roll), attesting to cultural belief in her powers of protection from peril [10]. However, on the whole there is very little surviving first-hand evidence from medieval women themselves, regarding either their obstetric treatment or the complications surrounding their own bodies [11,12]. The medieval Church offered a bevy of relics and talismans, in various forms, that were believed to be efficacious in bringing about a safe pregnancy and delivery. A supplicant might gaze upon, touch, kiss, wear, recite, venerate or even ingest this divine protection [13]. Midwives deployed parchment amulets, precious stones and plant-based remedies during childbirth [14]; the list of items that the church lent out to pregnant women is extensive. When Thomas Cromwell ordered the abbeys to be raided in 1536, in what would later come to be known as the Dissolution of the Monasteries, the raids were particularly vicious in targeting centres relating specifically to the veneration of female saints, much of which focused on childbearing and pregnancy [5]. The list of items seized included many items to aid childbirth. One such item was St Moodwyn's staff, which was lent to women in labour, 'to leane upon, and to walk with yt, and have greate confidence in the same staff' (to lean upon, and to walk with it, and have confidence in the same staff) [15, p. 384]. Others include the smoke of St Mary, or even her breast milk [16]. However, the most oft-recited item that a monastery lent out to its parishioners was a birthing girdle.

The material from which these girdles were made varies widely. The girdle of Our Lady from Bruton was made of red silk [17]. At Coverham, the girdle of Mary Nevell, good for women lying in, was made of iron [18]. Abbeys, such as Westminster, loaned or 'rented' out these girdles to women. For example, when Henry VII's wife fell pregnant, the sum of six shillings and eight pence was paid 'to a monke that brought our Lady gyrdelle to the Quene', noted in Elizabeth of York's Privy Purse Expenses, December 1502 [19, p. 78]. Girdles had several functions. They were to be worn and unloosed before marriages, and a similar 'girding and unloosening' occurred during pregnancy: girdles may have literally supported the extra pregnancy weight and then been loosened six weeks prior to confinement [20].

However, it seems that birth girdles, especially birthing rolls such as MS. 632, were often talismanic, with ritual functions that incorporated religious devotion and magic. Here, these artefacts formed part of a broader cultural landscape in which both women and men appealed to divine and supernatural forces for assistance in the face of ill health or danger. Devotional objects, such as relics or statues of saints, were venerated and touched in order to harness their beneficial power [21]. Religious gestures like these sat

alongside magical practices, most obviously the use of charms—formulae whose efficacy was derived from the power of words [13]. Women's health, encompassing not just pregnancy and birth but also menstruation and other gynaecological matters, alongside the various health issues shared with men (digestive complaints, problems with the eyes and ears, pestilential illness, etc.), was often addressed in the arena of oral culture, where religious and magical rituals were prominent. At the same time, a generous record of written material about women's health is extant from late medieval England, especially vernacular texts that may directly reflect oral culture (e.g. charms) [22,23]. Nonetheless, certain aspects of the social history and lived experience of women's health are difficult to uncover. Miracle accounts are a rich source of information about behaviours that may otherwise be undocumented, and include references to the use of garments or belts associated with saints to assist women in labour. An account from the 1170s of a healing miracle of St Thomas Becket describes how Alditha of Worth, after a labour lasting three days, was finally able to deliver her infant when encircled by a stole that St Thomas had blessed [24]. Here, as in other contexts, the emphasis is placed on the encircling role of the artefact, binding and protecting the maternal body.

Although this devotional and magical context is essential to understanding the cultural place of birth girdles, their use is also attested in medical texts, with occasional mentions in such texts of girding the loins of women in childbearing. The twelfth-century *Trotula*, a widely disseminated medieval manual on women's health, states that the skin of a snake should be hung about the abdomen. If a woman was having difficulty with a child coming out, 'let the woman be girded with a snake's skin from which the snake has emerged' [25, pp. 102–103]. *The Sickness of Women*, a fifteenth-century English vernacular medical text, also recommends a birth girdle. For the grievances of women travailing in childbirth, one remedy recommends, 'and lete guyrden hir with a guyrdel of an hertis skynne' (and let her be girded with a girdle of hart's [deer] skin) [26, p. 532]. *The Sickness of Women* is of especial relevance to MS. 632, since it evidences the later medieval birthing practices that were current at the time of the roll's production and, like the text on the roll, it is written in the vernacular, making it more accessible than Latin texts. These girdles, of silk, iron, parchment, or snake or deer skin, do appear to have been used by some pregnant women during the Middle Ages, especially by wealthy women, and were on the list of devotional items that were taken away during the Dissolution.

Perhaps because of the fervor with which the birthing talismans were destroyed by Reformers, or perhaps because post-medieval libraries favoured codices over rolls [21], or perhaps because of the delicate nature of the organic fibres that some other girdles were made from, few of these survive. The rolls that do survive, at least the ones that are known to us, are mainly made from parchment, such as MS. 632 (figure 1), although one printed on paper is also known (London, British Library, MS Harley 5919) [27]. In addition to MS. 632, there are seven other English girdles known to us in the British Library and other collections, and at least two French girdles also held by Wellcome Collection.

However, what makes MS. 632 largely unique among the remaining parchment rolls, is that it has obvious signs of actual use as a birth girdle. The images on the manuscript are particularly worn, especially those likely to be touched, rubbed or kissed as part of religious veneration, such as the nearly rubbed-out green crucifix, attesting to its use (figure 1) [21]. Several of its texts and crudely drawn images are associated with childbirth. It makes special invocation to Saints Quiricus and Julitta, mother and son martyrs, who in English birth roll traditions are invoked for aid specifically in childbirth [27]. The prayers in many of the rolls, MS. 632 included, are diverse—warding against many kinds of troubles—and owing to its various 'catch-all' nature of inclusion, the roll may have been used by men as well as women to guard against danger or hardship [28,29]. However, MS. 632 also crucially contains one final invocation specifically for women: '*And yf a woman travell wyth chylde gyrdes thys mesure abowte hyr wombe and she shall be delyvyrs wythowte parelle and the chylde shall have crystendome and the mother puryfycatyon*' [And if a woman travailing with child girds this measure about her womb, she shall be delivered safely without peril and the child shall be christened and the mother purified] [30, pp. 491–493]. Furthermore, the roll includes several invocations to the Virgin Mary, who was understood to provide much assistance to women in pregnancy and childbirth [24]. One of these references invokes the tradition of the letter to Pope Leo, in which an angel delivered a letter to the Pope from the Virgin herself, transferring particular rites of protection to '*who so beryth ths mesure uppon hym*' (*who so bears this measure upon him*). The 'mesure' was thought to be the height of the virgin and this height, common throughout protective rolls, was carefully reflected in the roll's length (330.0 cm × 10.0 cm). It appears that the precise measurement of the roll harnessed apotropaic powers.

Yet this was not the only item on the roll to invoke supernatural aid. The writing on the manuscript appropriates the precise numbers, invocations and repetitions found in magical talismans and

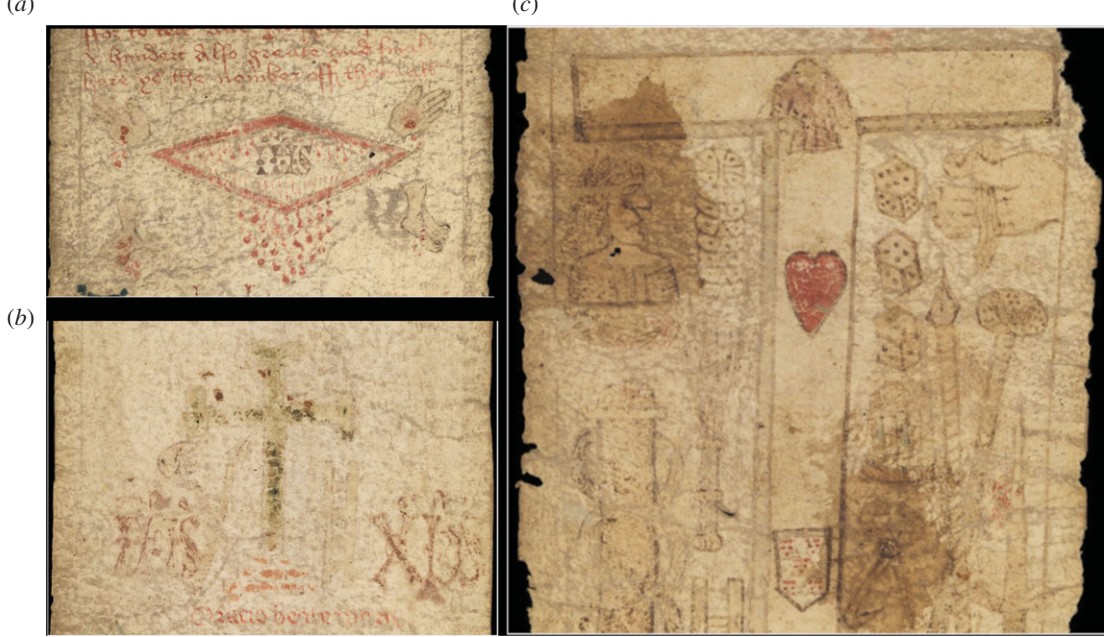

**Figure 1.** Details taken from MS. 632. (*a*) MS. 632: the dripping side-wound. (*b*) MS. 632: rubbed away green cross or crucifix. (*c*) MS. 632: Tau cross with red heart and shield. Images courtesy of Wellcome Collection.

enchantments into a Christian context. It includes the 12 names of the apostles, the names of the three magi, the repetition of precise numbers (the exact number of the drops of Christ's blood) and the use of sacred holy names, both in invocation and in abbreviation (such as the holy initials of Christ). These names were used to invoke protective powers. The protective powers were also invoked in the dense block of text, which Rudy as suggests 'may have served as a hermetic net for evil and disease' [13,21, p. 45]. The precise measurement of the roll, the careful numerization of sacred names, the block texts and holy images worked collectively to fend off all manner of harm—from death in battle to childbirth [21]. Such features indicate that the highly formulaic rituals blend magic rituals with religious protection [13].

MS. 632's severe abrasions implies that it was often touched or kissed, and accords with widespread evidence of medieval votive practices, where an image was kissed or rubbed so frequently the image is worn and blurred [21]. Its narrow width (330.0 × 10.0 cm) suggests that it was intended to imitate an actual metal or cloth girdle that could be wrapped around a woman's body, with the strategic placement of particular prayers against her womb [27]. The folds in the parchment of MS. 632 certainly attest to such a type of arrangement and the length of the manuscript would make this possible. If the roll was wrapped around the waist, draped down the backside, lifted up through the legs and draped back over the woman's abdomen toward her chest—this would fit the cross-like shape suggested by Gwara, Morse, Jones and Olsan. Figure 2 demonstrates various wrapping techniques that could, plausibly, encircle and completely gird a woman with the roll during labour.

It is worth noting that MS. 632, at its earliest, was created in England during the late fifteenth century. The Dissolution of the Monasteries began in 1536 and carried on until roughly 1541 [10]. The window of the use of MS. 632, between its creation and the Dissolution, in theory, was only 60 years. The evidence of Dissolution destruction points to targeting birth girdles specifically [10]. However, MS. 632's mere survival is testament that it was not destroyed during the Dissolution. Protestant Reformers were quick to target the rituals of childbirth as these were seens as potential 'sanctuaries for forbidden religious practices' [31, p. 67]. Protestant bishops specifically banned birth girdles, and even forced midwives to take oaths to say that they would not employ them [31]. Yet, Cressy and Thomas both note that recalcitrant midwives frequently employed birthing girdles on the sly in post-Reformation England [32,33]. Early modern practices, such as those of Thomas Lupton in 1579, encouraged encircling the womb with a snake-skin [31]—a practice employed in the medieval *Trotula*. However, during the Reformation, power fluctuated between Protestant and Catholic monarchs, with each monarch also influencing birthing practices [31]. While certain medieval items, such as the Eagle Stone, increased in popularity as a birthing remedy in Early Modern England, birth girdles, over time,

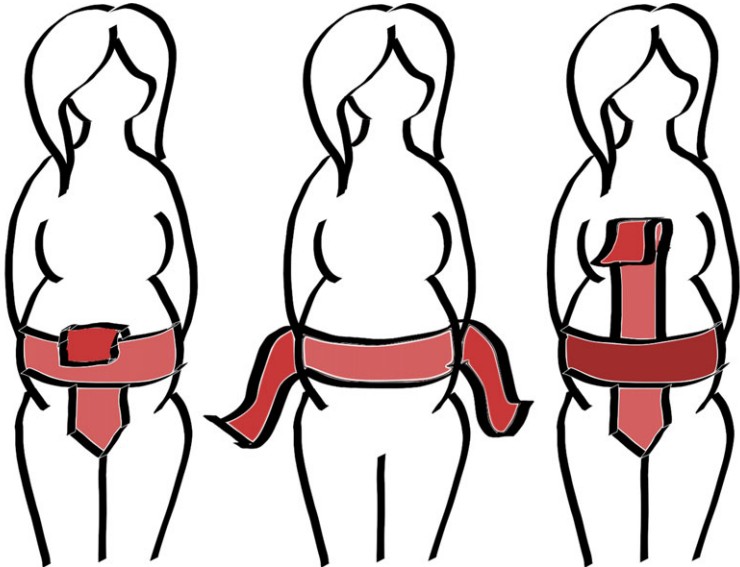

**Figure 2.** Three possible methods of tying the birth girdle when used pre and during labour.

were repeatedly prohibited [31,34]. It is difficult to tell how long MS. 632 was in use: was it stored quietly away during the Dissolution? Was it one of the birth girdles being used in secret against post-Reformation authorities? Clandestine retention and use of the roll beyond 1536, even for some time, is certainly not impossible, especially given its powerful protective functions. No information survives about the early owners of the roll to help to inform us about its use. While we cannot ascertain the girdle's exact length of active use, its chronological window reflects shifting systems of power, frequently male-exerted, that shaped women's childbearing experience [35].

The overall effect of the manuscript is that it has been very heavily worn, often thought through use, with the significant abrasions on the surface of the parchment making large sections of the text illegible. The abrasions most probably indicate that it has frequently been used as a birth girdle [30]. As many of these 'birth girdles' invoke protection for multiple uses, such as protection in battle, their use in child-birth has been alleged, even called 'so-called birth girdles' [10]. But Skemer argues that the abrasions on these girdles testify to their active use in labour [13]. The devotional images on MS. 632, such as the cross, are particularly worn which aligns perfectly with votive use in other rolls and devotional manuscripts outlined by Rudy [21]. In fact, the heavy abrasions and wear of MS. 632 has even called for it to be tested to see if its use in childbirth can be scientifically verified. Lea Olsan writes: 'Were they [birthing rolls] damaged during repeated use in the events surrounding labour and delivery, when the roll, according to its instructions, was laid over the womb of the woman to ease the delivery? There are a few reddish marks that could be blood stains where the roll is very worn, but laboratory work will be required to make certain' [36]. In this work, we take up Olsan's challenge and attempt to provide identification of the stains on these samples.

## 1.2. Recent advances in palaeoproteomics and non-invasive techniques

Recent advances in palaeoproteomic techniques have allowed a much more in-depth examination of biomolecular evidence on different substrates. Proteomic analysis of bone [37–39], tooth enamel [40,41], shell [42] and mummified skin tissue [43] have proved the survival of a diverse collection of proteins in addition to the primary structural protein. Proteomic analysis has also been successfully carried out on more recent objects like artwork [44,45], textiles [46] and manuscripts [47,48] with equally interesting outcomes. However, one of the obstacles to overcome is an inherent bias to the predominant protein present (in the case of parchment that protein is collagen) and consequently the signal of lower concentration proteins is drowned out, in most cases to such a degree that they are undetectable.

In addition, a further obstacle to access has been the method of sampling. Initial proteomic studies of cultural heritage objects have always required taking physical, destructive samples [47], severely limiting the amount of objects that can be subjected to this kind of analysis. Recently, there has been a move to less invasive forms of sampling. The use of non-invasive ethyl vinyl acetate diskettes on numerous paper documents [48–50] and even mummified skin [51] has provided interesting details about the use of these

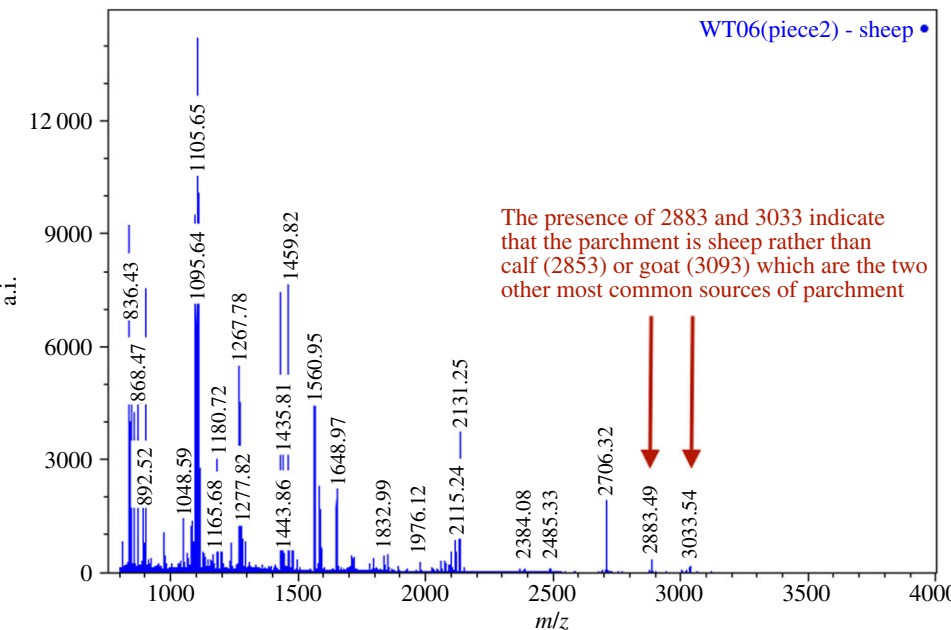

**Figure 3.** Example of MALDI-TOF spectra from one of the birth girdle samples (WT06) identified as sheep parchment.

objects, and, in some cases, possible evidence of pathogens has been detected. It has recently been used for the first time on parchment [52], however, owing to the nature of the sampling (requiring the damp diskette to be in contact with the parchment), it may limit its widespread adoption. Here, we report, to our knowledge, the first proteomic analysis of a historic parchment document using a dry non-invasive sampling technique (eZooMS). This technique was developed for use on parchment documents allowing for the extraction not only of collagen [53] but also of DNA both from the animal and the microbiome [54]. This sampling technique has now been used to analyse the broader set of proteins present on the surface of the document which can provide information about the history and use of this object.

## 1.3. Samples

Made in England, Wellcome MS. 632 is a long and thin roll, dating to the late fifteenth or early sixteenth century, measuring 330.0 cm × 10.0 cm. It is made of four strips of sheepskin parchment stitched together. The roll contains: (i) the three nails of the crucifixion; (ii) the crucifix with a heart and a shield; (iii) a circle surrounding the Holy monogramme, 'IHS' (the first three letters of the name 'Jesus' in Greek–iota (i), eta (h), sigma (s)); (iv) the hands and feet of Christ (i.e. the five wounds of Christ) dripping with blood; (v) a very rubbed crucifix, faded; and (vi) a diamond-shaped figure; very badly rubbed, possibly Christ standing [28,29,30]. A unique point relating to this girdle is that it has text on both the face (recto) and dorse (verso) of the roll, which is unusual.

Eight eraser samples were taken from Wellcome MS. 632, including one visually 'unstained' area (electronic supplementary material figures S1 and S2). At least one sample came from each of the four different pieces of parchment that make up the roll in order to determine the species of animal used to make the parchment in each of the cases.

# 2. Results and discussion

## 2.1. eZooMS

All eight eraser samples were analysed via matrix-assisted laser desorption/ionization-time of flight (MALDI-TOF) analysis for species identification. In all cases, the species of parchment was determined to be sheep (figure 3). We can conclude, therefore, that all four separate pieces of parchment that make up the birth roll are all made from sheep parchment. Previous studies have shown a predominance of sheep parchment in English legal documents [53]; however, in this case the reason for using sheepskin may be for more practical reasons. Given that sheepskin is more affordable than calfskin, and more readily available in England than goat, it would seem an obvious choice.

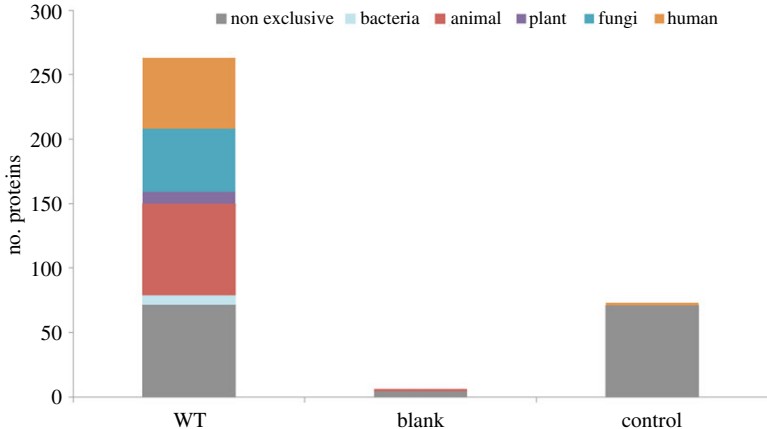

**Figure 4.** Shared proteins and exclusive proteins by category found on the three groups of samples analysed: birth girdle (WT), blank (paper) and control (eighteenth-century parchment).

More importantly, sheepskin is much thinner than calfskin, lending itself to make documents that need to be folded or manipulated, such as in the case of the birth girdle.

## 2.2. Liquid chromatography-tandem mass spectrometry

A list of proteins found in the birth roll samples can be found in the electronic supplementary material, table S1. Of the 273 proteins found in the birth roll samples, 200 were found exclusively in these samples (figure 4). Only proteins that were found exclusively in the birth roll samples are discussed in the results. Differential expression, therefore, was not taken into account in this study (merely presence/absence), but there is the potential for these studies to be conducted at a later stage.

## 2.3. Non-human proteins

> 'Take one scruple of opium poppy, one scrupe of goose fat, four scruples each of wax and honey, one ounce of oil, the whites of two eggs, and the milk of a woman. Let these be mixed together and inserted by means of a pessary', (Trotula, [25, p. 89]).

We were able to detect the presence of a number of animal-derived proteins including honey (royal jelly protein) in WT04, ovicapra milk-derived products in WT07 (casein), egg white (ovalbumin) and egg yolk (although two peptides also found in the control). We were also able to detect the presence of leguminous plants (broad beans and possibly garden pea), as well as cereals (figure 5).

### 2.3.1. Honey

One of the birth girdle samples (WT04) presented three peptides (IMNANVNELILNTR, LLTFDLTTSQLLK, MVNNDFNFDDVNFR) identified exclusively to the Major royal jelly protein 1 (MRJP1, ID:O18330) from *Apis mellifera* (honeybee).

Honey has been used throughout history for medicinal purposes, both for its natural antiseptic properties as well as its sweetness to make mixtures more palatable. In the fifteenth-century obstetric Middle English text, *The Sickness of Women* honey is frequently employed [26]. It is used as an ingredient in suppositories and to anoint the womb before the entry of suppositories. Honey was used in recipes to cure heavy menstrual flows; to make plasters; to remedy uterine prolapse; and to purge the womb of bad humours or inflammation. Honey was used as an expedient to deliver a child, as well as to help expel a dead fetus from the womb. For example, it was mixed with ingredients such as myrrh, castory and calamint, to ease delivery [26]. It was used in cures to expel uterine moles and uterine ulcers.

Influential for centuries after its composition in the twelfth-century, the *Trotula* also used honey to remedy problems in childbirth, including for movement of the womb from its place. For lesions of the womb, ptisan and honey were combined and inserted into the womb as a pessary [25]. Honey again is used on poultices to tie up the womb after it has come out after birth. Pulverized herbs (garden cress, laurel berries, frankincense and cinnamon) mixed with honey were placed on the loins and tied with a band in a ball, and inserted into the vagina [25].

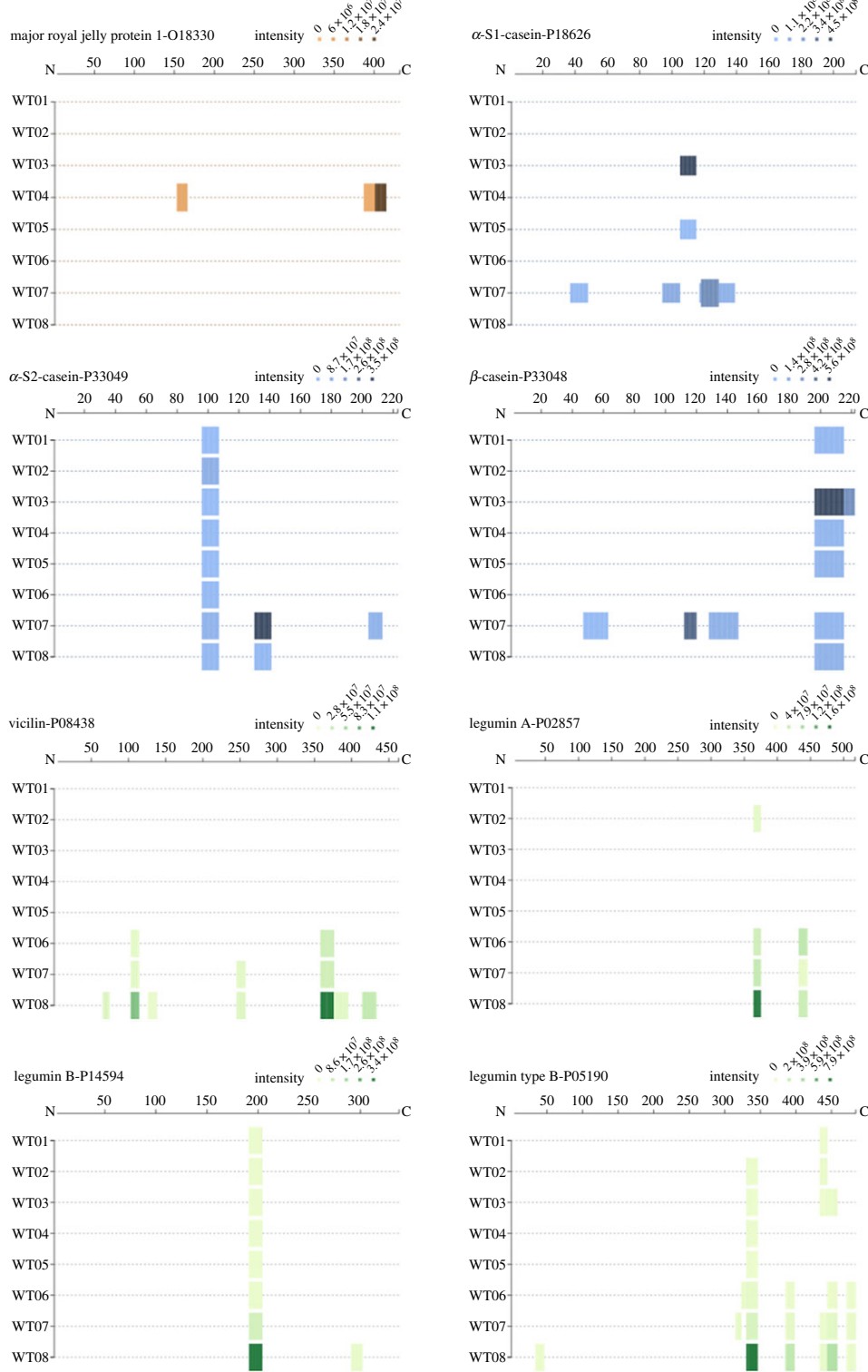

**Figure 5.** Plots showing all the peptides present for the main non-human proteins detected in the birth girdle samples using PEPTIGRAM [63]. Each coloured band represents one peptide sequence and the intensity of colour reflects the peptide intensity detected.

### 2.3.2. Egg

Vitilogenin-2, a protein found in egg yolk was detected in two of the eight birth roll samples. However, two (different) peptides were also observed in our control parchment and therefore we cannot exclude some form of contamination. However, two peptides from Vitilogenin-1 were exclusively found in the birth roll samples, making a stronger case for the presence of egg yolk.

There is evidence of the use of egg yolk during childbirth, and the complications surrounding women's health, in a medieval treatise contemporary to the dating of MS. 632. In *The Sickness of Women*, egg yolks were used in recipes to promote conception [26]. Similarly, an earlier text, the *Trotula*, recommended that to avoid a difficult birth the mother should eat easily digestible foods for the last three months of pregnancy, including yolks of egg [25]. Early modern textbooks, probably reflecting earlier medieval practices, employed egg yolks as lubricant to facilitate delivery in the case of a difficult labour [55–58]. Another recommends that birth mothers should eat certains foods to keep their strength up during labour, including a poached egg yolk [55,58,59]. Egg yolks, from medieval to early modern practices, habitually occur in birthing practices.

The presence of egg white was detected in all eight birth girdle samples but not in the blank or control. Egg white is often used to treat the surface of the parchment [60], so in the case of the birth roll we cannot conclude categorically that its presence is owing to medicinal purposes as it could equally be explained by virtue of the preparation of the parchment. In addition, chicken lysozyme is detected in many of the same samples (lysozyme is also found in egg white).

Egg whites are also found in medieval recipes for women's health. In *Sickness*, egg whites were used in recipes to cure the womb's swelling or to purge the womb of bad humours during pregnancy [26]. Egg whites were used as ingredients in suppositories and in cures to wounds in the womb caused by childbearing [26]. One reference in the *Trotula* regards the use of egg whites for ulcers of the womb, thought to be occasioned by miscarriage. To mitigate the pain, the text indicates that one should apply the 'juice of a deadly nightshade, great plantain with rose oil, and white of egg with woman's milk and with purslane juice and lettuce,' [25, p. 93]. Another recipe regarding 'excessive heat of the womb' uses two of the items found on the Wellcome birth girdle. 'Excessive heat of the womb' draws on the Hippocratic of bodily humours: women were normally cold and moist, but pregnancy could upset this balance. If the womb was thought to be too hot, this remedy was applied: 'Take one scruple of juice of opium poppy, one scrupe of goose fat, four scruples each of wax and honey, one ounce of oil, the whites of two eggs, and the milk of a woman. Let these be mixed together and inserted by means of a pessary' [25, p. 89]. This uses honey and egg whites, both found in the analysis.

### 2.3.3. Milk

Ovicaprine (sheep or goat) milk proteins were predominantly detected in WT07 (although certain peptides were also present in other samples) including: alpha-casein 1 (five specific peptides), alpha-casein 2 (two peptides) and beta-casein (five peptides). Milk has been described as a surface treatment of parchment [60,61]; however, we found no evidence of milk on the control parchment which would seem to indicate that the presence is owing more to the nature and use of the birth girdle. It is interesting to note, unlike beta lactoglobulin (BLG), the milk protein most commonly reported in dental calculus, that the only milk protein detected was casein. In separated milk, BLG is enriched in the whey, while casein is concentrated in curds, and therefore also present in cheese and other milk-derived products, with any one of these being the possible origin.

Goats' milk was commonly used in child birthing remedies. In *Sickness*, goat's milk was used to 'helpen to have furth a ded chield from the matrice' [help to bring forth a dead child from the womb] [26, p. 552]. Galbanum was to be 'resolued in gotis mylke that the womman may vse easily' (resolved in goat's milk that the woman may use easily) [26]. Goats' milk was consumed to strengthen women after blood loss, a frequent occurrence in childbirth, thought to thicken the blood, heal varicose veins and prevent the blood from flowing [26]. Similar remedies included consuming linseed steeped in either sheep or goats milk [26]. It is also used in recipes on windiness in the womb [26]. Moreover, cheese was used a consumable amulet or pessary in childbirth rituals. For example, the *Trotula* instructs writing symbols on cheese or butter and giving them to the mother to eat, in order to hasten a delivery [25,62], which could offer a possible explanation for the presence of casein on the birth girdle.

### 2.3.4. Plants

#### 2.3.4.1. Leguminous plants
We detected seven peptides from vicilin from *Vicia faba* (broad bean) in three of the birth girdle samples. In addition, we also detected the presence of peptides from two other proteins, legumin A (two peptides) and legumin B/type B (10 peptides) although the specificity to the organism cannot be differentiated between broad bean, garden peas and common vetch.

Leguminous plants were frequently used in medieval recipes, including those for women's health. For example, *The Trotula* uses beans for lesions of the womb, to instigate the flowing of breast milk and for curing certain pregnancy cravings [25]. Some of these recipes include mixing the bean with wheat and barley—which too are detected in this manuscript. *The Sickness of Women* uses bean meal as a dietary remedy for dropsy and for swollen legs during pregnancy; beans and peas were used to increase fertility in both men and women [26]. Considering legumes were used in medieval recipes for childbearing, their presence in MS. 632 may be explained in terms of medieval remedies surrounding childbirth. The legumes' presence may even suggest what types of ingredients were used in England as remedies for problems surrounding childbearing. While medieval texts do not provide information about when birth girdles were removed from women's bodies after childbirth, there is also the possibility that the legumes reflect the restorative dietary provision that women received postpartum.

### 2.3.4.2. Cereals

We were also able to detect the presence of cereals in all the birth girdle samples (although mostly in sample WT02), but many of the peptides were not specific enough to determine the exact species, the closest matches included wheat, barley and spelt. There is the possibility that their presence could be explained by the use of a conservation treatment using wheat or flour paste. However, it would be unlikely that they used all of the cereals and we have yet to find documentary evidence that any conservation of this kind has been undertaken. Cereals can also be used as a surface treatment in parchment production [60] and could, therefore, explain their presence. The roll, however, does not appear to bear any signs of conservation work, including the use of wheat starch paste, indicating that the present of wheat is historical, probably accumulated through use.

There is ample evidence of cereals, including wheat and barley, used for treatments surrounding women's health and childbirth. Barley was a particularly versatile ingredient, used in remedies for excessive menstrual blood flow, menstrual pain, lesions of the womb, vaginal wind, the expedition of a delivery and as a contraceptive. *Sickness* employs barley meal as key ingredient for a plasters—often employed for child birthing remedies (inflamed womb or varicose veins) [26]. Wheat bran was combined with the urine of each couple to determine causes of infertility [26]. Wheat baths remedied varicose veins [26]. Wheat-meal was used for plasters for inflamed wombs: 'take whete mele and oile of olive and hony and make therof a plaster' (take wheat meal and oil of olive and make thereof a plaster) [26, p. 519]. When a woman's womb caused pain after being displaced in childbirth, the remedy was to feed her cakes made of elder, and mixed with eggs and wheat to make thin cakes, fried in grease [26].

For difficult deliveries, the *Trotula* instructs a woman to be bathed in water in which barley, mallow, fenugreek and linseed have been cooked. In another instance in the *Trotula*, if a woman was badly torn and did not wish to give birth again, she must put barley or grains of caper spurge into her afterbirth, the amount of grain for the number of years she wished to remain barren. Wheat flour was used to stimulate a maternal milk-flow, to treat lesions of the womb and to determine male or female cause of infertility [25]. While the presence of cereals on the birth girdle may indicate parchment preservation, it may also indicate their active use in treatment surrounding childbirth, thereby affirming evidence that such recipes as present in the *Trotula* were widely used.

All of these products—honey, eggs, milk, beans, peas, cereals—were part and parcel of the daily life of medieval Britain. These ingredients provided daily sustenance, even post-partum nutrition. While the birth girdle could have easily come into contact with any of these ingredients in numerous ways, it is interesting to note that every single one of these ingredients was used to facilitate medieval childbirth, frequently in combination with one another. The veneration of holy objects, such as MS. 632, indicates that it was probably treated with reverence when not in active use [21]. Such veneration makes it less likely that the girdle would have acquired a cereal protein from a monk's breakfast in the Refectory and more likely that these ingredients accumulated on the manuscript through its daily functions—one of which, we demonstrate, was probably childbirth.

### 2.3.5. Human proteins

A total of 55 human proteins were found exclusively in the birth roll samples, compared to two in the control sample (electronic supplementary material, table S2). These proteins were then compared to the proteome from cervico-vaginal fluid (CVF) published by Muytjens *et al.* [64].

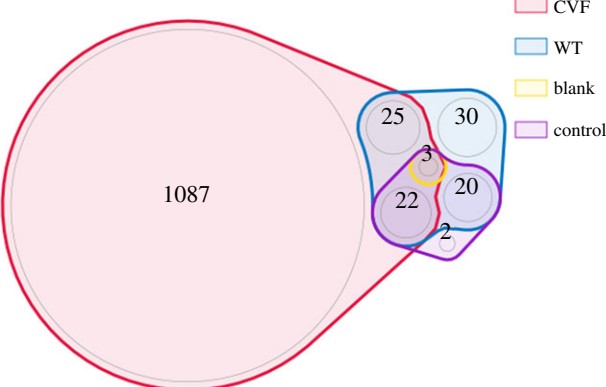

**Figure 6.** Venn diagram of human proteins present in the different samples compared to proteins found in cervico-vaginal fluid (CVF). WT (birth girdle) samples have the largest number of human proteins, and the largest number of exclusive proteins found in CVF, compared to the blank and control.

A higher number of proteins from the birth roll samples are detected in CVF than from any other of the sample groups. This can provide a further plausible indication that the roll was indeed actively used during childbirth (figure 6).

As detailed above, the assumption about the method of use of birth rolls is that they were worn during childbirth by the mother, possibly tied around the waist like a girdle, acting as a talisman or good luck charm for what has always been a significantly dangerous and potentially fatal experience for women. If indeed the roll was used in this way, it would be logical to expect to find traces of bodily fluids and medical remedies present on the roll.

In addition, we were also able to detect nine peptides belonging to major urinary proteins from *Mus musculus* (mouse) which would indicate that during the history of the object, at some point it has been stored in place where mice have had access to it and left this evidence. However, we cannot say when this happened. In addition, we recorded a large number of peptides belonging to different strains of *Aspergillus*, something that has also been noted by other researchers looking at parchment [52]. *Aspergillus* seems to often be found colonizing parchment samples, but to what degree this may be a conservation concern requires further research.

The chronological window during which these proteins embedded on MS. 632 is cautiously suggested between 1475 and 1625. The birth girdle dates to the late fifteenth-century, at earliest, providing the first date. The end date is more difficult to determine. In sixteenth-century England, power fluctuated between Catholic and Protestant monarchs, each influencing birthing practices, including the use of birthing girdles [30]. Because birth girdles were targeted during the Dissolution, this girdle may have been quietly stored. If this is the case, the girdle's active years of use extend from roughly 1475 to 1536—only 60 years of use. Despite the ban on birth girdles by bishops, women were known to still use birthing girdles of the Virgin up until the late sixteenth and early seventeenth centuries [31–33]. Given that this girdle was not destroyed by Reformers and given the heavy wear of MS. 632, it may be that this was a clandestine girdle used up until the early seventeenth century. The dates of MS. 632's use indicate that the window for the proteins to be deposited could span between 60 and 150 years.

Remembering the words of Lea Olsan cited in the introduction, obtaining direct evidence of use of these birth girdles is almost impossible save for the idea that direct biomolecular data could be extracted and identified in a laboratory setting. We have now shown that with current non-invasive methodologies it is possible to obtain direct evidence of use that provides clear data to support the historical scholarship of birth girdles.

## 3. Conclusion

Non-invasive proteomic analysis of manuscripts can provide a deeper understanding of the history and use of the object. Surface sampling preferentially extracts substances that have been deposited on the document and avoids the inherent bias from predominant proteins when using physical destructive samples. Proteomic analysis has the advantage over genetic analysis in that it is tissue-specific providing information not only on species but materials used. We have demonstrated its applicability with the example of this medieval birth roll providing proteomic evidence of its possible active use

during childbirth. We have also proved what was long suspected of MS. 632, that its badly worn state attests to its use during childbirth. The use of the honey, legumes, eggs and even milk products are also common remedies in childbirt, and this study lends further support to medieval medical treatises, such as those recorded in obstetric manuals, were practices that were actively employed. The active use of MS. 632 in childbirth also shows that women were using highly formulaic rituals that blended the numerical precision and incantation of magic with religious protection. The potential of proteomic analysis applied to the vast corpus of parchment documents represents a huge new avenue of exploration for the burgeoning field of biocodicology.

# 4. Material and methods

Samples were taken using PVC erasers as described in Fiddyment *et al*. [53]. Samples were extracted using conventional eZooMS methodology [53] and initially analysed using MALDI-TOF MS. Eraser crumbs collected from rubbing the eraser on a blank sheet of paper acted as a blank, and rubbings taken from an eighteenth-century Scottish legal document acted as a control.

## 4.1. Liquid chromatography-tandem mass spectrometry

The remaining peptides left from the eZooMS analysis were dried down in an evaporator and sent for analysis to the University of Copenhagen (Globe Institute and Centre for Protein Research) where they were analysed via liquid chromatograph-tandem mass spectrometry (LC-MS/MS) following the Copenhagen protocol from Demarchi *et al*. [42].

## 4.2. Data analysis

Data were analysed using the open source software MAXQUANT [65] (v.1.6.10.43) with tandem mass spectra searched against the Uniprot_SwissProt database (downloaded 23 January 2020) including in the search all common contaminants (cRAP). Mass tolerance was set to ±5 ppm on the precursor and 0.05 Da on the fragments. The thresholds for peptide and protein identification were set as follows: protein false discovery rate = 0.01, protein score and peptides matches greater than or equal to 1. Subsequent analysis was then carried out using the Proteus R package (https://github.com/bartongroup/Proteus) [66]. Only proteins with greater than or equal to 2 peptides were considered during the results and analysis.

Data accessibility. The mass spectrometry proteomics data have been deposited to the ProteomeXchange Consortium via the PRIDE partner repository [67] with the dataset identifier PXD022054.

Authors' contribution. S.S. and S.F. came up with the initial question and S.S. undertook the non-destructive sampling. S.F. extracted and analysed all the samples. S.F. and M.J.C. interpreted all the data. E.B., N.J.G. and K.P. conducted all the historical contextualisation. All authors gave final approval for publication and agree to be held accountable for the work performed therein.

Competing interests. We declare we have no competing interests.

Funding. This work was supported by ERC investigator grant no. 787282-B2C. S.F. was additionally supported by British Academy Postdoctoral Fellowship funding. M.J.C. acknowledges support from the Danish National Research Foundation DNRF128. K.P.'s research is supported by a CHASE Doctoral Training Partnership.

Acknowledgements. The authors would like to thank Meaghan Mackie at The Globe Institute, the University of Copenhagen for her help with running the LC-MS/MS samples and her advice on data preparation. The authors would also like to thank Durham University's Institute of Medieval and Early Modern Studies for facilitating a portion of this collaboration.

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
