## [Peer Review File · Royal Society Open Science]

Revised submission:	8 January 2021	
Final acceptance:	3 February 2021	

Review History

RSOS-202055.R0 (Original submission)

Review form: Reviewer 1

Is the manuscript scientifically sound in its present form?

No

Are the interpretations and conclusions justified by the results?

Yes

Is the language acceptable?

Yes

Do you have any ethical concerns with this paper?

No

Have you any concerns about statistical analyses in this paper?

No

Recommendation?

Accept with minor revision (please list in comments)

Comments to the Author(s)

The manuscript "Girding the loins? Direct evidence of the use of a medieval parchment birthing girdle from biomolecular analysis" describes the proteomics results from the analysis of eight samples taken from a birth girdle composed of four parchments and stained from what the study demonstrates as possible cervico-vaginal fluid resulting from the use of the girdle during birth. A large array of other non-human proteins were also identified, providing evidence of ways that the girdle was used. The subject of this study is very interesting and original, and the study shows the potential of biomolecular methods, and especially proteomics here, to uncover the use history of such rare objects. The subject is well documented, which makes for a very interesting read and brings the background history of these objects into light. The presentation of the results, however, seems rushed and incomplete and could benefit from a more detailed description. More details of the analysis and results could be given in Supplementary Materials, such as detailed tables of peptide and protein identification. There is no comparison of the stained and non-stained spots, so it is not clear whether the human proteins were found in these locations only.

Detailed comments:

Abstract: It needs re-writing, as it insists on the potential of the method more than on the originality of the case study which is the real focus of the paper. "Dry non-invasive sampling method" is mentioned three times in the abstract, yet this paper is not about the description of a new method. In that regards, the first sentence is misleading. In fact the methods are not described at all, even in Materials and Methods, and instead refer to two published papers. Either the innovative aspect of the extraction, if any, needs to be detailed, or the abstract should clarify that the focus is here rather on the application of already developed methods, and highlights the importance of the results instead. Furthermore, the study is about one manuscript and does not represent a large scale study of "stained manuscripts". While the method could potentially work on all blood-stained parchments, this is an extrapolation that doesn't take into account the possible special preservation circumstances of the studied manuscript (such as heavy and/or repetitive use), which is a point that was not really addressed in the manuscript. "both human and non-human peptides from the stains including evidence for the use of honey, cereals, ovicaprine milk and legumes": in the paper you mention that both stained and non-stained samples were taken, but did not give further detail on which samples proteins were found. Does that mean all these proteins were found in stained areas?

Keywords: birth and stains are very generic. Perhaps gives more precision: birth girdle, blood stains...

Introduction: starting the paragraph with "even today" sounds odd. It feels like the sentence should be reversed. The references on modern birth are of little use here unless you want to make a direct comparison.

Line 6 to 8: Even repeated twice and makes the sentence heavy

Line 26: "siezed" instead of seized

Page 7: the Samples paragraph, as well as the first two sentences of eZooMS from Materials and Methods should be moved at the end of introduction as they contain information to understand the results and a valuable description of the object. Furthermore a sort of map showing where the samples were taken in relation to the stains should be added.

Page 7 and 8: the two paragraphs about proteomics and non-invasive techniques should be merged into one. Lines 27 to 30 should be linked to lines 9 to 12 since you come back to the idea of identifying proteins besides collagen, while the rest of the paragraph (line 30 to line 4) seems out of context and irrelevant here and could be deleted. On the other end, lines 7 to 9 refer to the Non-invasive techniques paragraph, so should be written together.

This paragraph could be developed as well to provide more references about analysis on non-collagenous proteins. Given the proteins identified in this study, work done on ceramics, mummies (for ex: Corthals et al, 2012; Jones et al, 2016) or artwork (numerous studies on paintings with egg and milk binders, work on animal membranes Popowich et al, 2018) would be more appropriate references.

Please provide references on the development of your eZooMS technique and your own work on parchments to give more context to the approach taken here.

“The technique has now been expanded”: did that require any technical improvement?

Results:

P 8, L 27: Did the LC-MS/MS confirm that identification and did you identify collagen from another species in LC-MS/MS that could come from animal glue?

Figure 3: Indicate markers that allow identification of sheep, and/or a sheep reference spectrum
SM Table 1: Table difficult to read, with no logical order. Are the proteins classified in terms of frequency, score?

LC-MS/MS results: P9, Line 8 “Differential expression therefore was not taken into account in this study (merely presence/absence)”. The manuscript would benefit from such analysis. As far as non-human proteins are concerned, peptide tables should be provided with details of the peptide identification for each category of proteins and some MS/MS spectra (for ex for honey). For human proteins, a general protein table would help to show where these proteins were identified in relation to the sampling.

Egg peptides should be added in Figure 5 too.

P11, L14: “miscarraige” instead of miscarriage

P11: Milk. You say milk was found in WT07 but figure 5 indicates casein in multiple samples.

P11, L41: what is the difference between Legumin B and Legumin type B? Also re-order figures in Fig 5 to logically follow the order referenced in text

P12: cereals. In which samples?

Figure 5: in legend, indicate full names of proteins and species

P14: Human proteins. Which samples? Were the proteins more abundant in stained samples, compared to non-stained? Are there proteins that are specific to CVF?

Table 1: “48” from the Venn diagram, I count 47; “54” from the Venn diagram, I count 55

Furthermore Table 1 is redundant with the venn diagram and can be eliminated. The first two columns are already mentioned in the text

Control sample: explain your choice of an 18th c parchment for control. It would have been more appropriate to use a modern reference of sheep parchment without its own use history.

Conclusion:

L24: “Surface sampling preferentially extracts substances that have been deposited on the document and avoids the inherent bias from predominant proteins when using physical destructive samples”. Do you have data comparing the different sampling methods to support this? Surface sampling has the advantage of being less destructive and in such cases the only acceptable sampling method, but you could be missing proteins that have been absorbed in the parchment, or, as in the case of this object, evidence of molecules from earlier uses. Filtration and separation techniques with LC-MS/MS are efficient enough to be able to separate the predominant proteins from minor ones.

Review form: Reviewer 2 (Monica Green)

Is the manuscript scientifically sound in its present form?

Yes

Are the interpretations and conclusions justified by the results?

Yes

Is the language acceptable?

Yes

Do you have any ethical concerns with this paper?

No

Have you any concerns about statistical analyses in this paper?

No

Recommendation?

Accept with minor revision (please list in comments)

Comments to the Author(s)

Please see attached file (Appendix A).

Decision letter (RSOS-202055.R0)

Dear Dr Fiddymment

On behalf of the Editors, we are pleased to inform you that your Manuscript RSOS-202055 "Girding the loins? Direct evidence of the use of a medieval parchment birthing girdle from biomolecular analysis" has been accepted for publication in Royal Society Open Science subject to minor revision in accordance with the referees' reports. Please find the referees' comments along with any feedback from the Editors below my signature.

Please submit your revised manuscript and required files (see below) no later than 7 days from today's (ie 10-Dec-2020) date. Note: the ScholarOne system will 'lock' if submission of the revision is attempted 7 or more days after the deadline. If you do not think you will be able to meet this deadline please contact the editorial office immediately.

Please note article processing charges apply to papers accepted for publication in Royal Society Open Science (<https://royalsocietypublishing.org/rsos/charges>). Charges will also apply to papers transferred to the journal from other Royal Society Publishing journals, as well as papers

submitted as part of our collaboration with the Royal Society of Chemistry (<https://royalsocietypublishing.org/rsos/chemistry>). Fee waivers are available but must be requested when you submit your revision (<https://royalsocietypublishing.org/rsos/waivers>).

on behalf of Professor Malcolm White (Subject Editor)
openscience@royalsociety.org

Reviewer comments to Author:
Reviewer: 1
Comments to the Author(s)

The manuscript “Girding the loins? Direct evidence of the use of a medieval parchment birthing girdle from biomolecular analysis” describes the proteomics results from the analysis of eight samples taken from a birth girdle composed of four parchments and stained from what the study demonstrates as possible cervico-vaginal fluid resulting from the use of the girdle during birth. A large array of other non-human proteins were also identified, providing evidence of ways that the girdle was used. The subject of this study is very interesting and original, and the study shows the potential of biomolecular methods, and especially proteomics here, to uncover the use history of such rare objects. The subject is well documented, which makes for a very interesting read and brings the background history of these objects into light. The presentation of the results, however, seems rushed and incomplete and could benefit from a more detailed description. More details of the analysis and results could be given in Supplementary Materials, such as detailed tables of peptide and protein identification. There is no comparison of the stained and non-stained spots, so it is not clear whether the human proteins were found in these locations only.

Detailed comments:

Abstract: It needs re-writing, as it insists on the potential of the method more than on the originality of the case study which is the real focus of the paper. “Dry non-invasive sampling method” is mentioned three times in the abstract, yet this paper is not about the description of a new method. In that regards, the first sentence is misleading. In fact the methods are not described at all, even in Materials and Methods, and instead refer to two published papers. Either the innovative aspect of the extraction, if any, needs to be detailed, or the abstract should clarify that the focus is here rather on the application of already developed methods, and highlights the importance of the results instead. Furthermore, the study is about one manuscript and does not represent a large scale study of “stained manuscripts”. While the method could potentially work on all blood-stained parchments, this is an extrapolation that doesn't take into account the possible special preservation circumstances of the studied manuscript (such as heavy and/or repetitive use), which is a point that was not really addressed in the manuscript. “both human and non-human peptides from the stains including evidence for the use of honey, cereals, ovicaprine milk and legumes”: in the paper you mention that both stained and non-stained samples were taken, but did not give further detail on which samples proteins were found. Does that mean all these proteins were found in stained areas?

Keywords: birth and stains are very generic. Perhaps gives more precision: birth girdle, blood stains...

Introduction: starting the paragraph with "even today" sounds odd. It feels like the sentence should be reversed. The references on modern birth are of little use here unless you want to make a direct comparison.

Line 6 to 8: Even repeated twice and makes the sentence heavy

Line 26: "siezed" instead of seized

Page 7: the Samples paragraph, as well as the first two sentences of eZooMS from Materials and Methods should be moved at the end of introduction as they contain information to understand the results and a valuable description of the object. Furthermore a sort of map showing where the samples were taken in relation to the stains should be added.

Page 7 and 8: the two paragraphs about proteomics and non-invasive techniques should be merged into one. Lines 27 to 30 should be linked to lines 9 to 12 since you come back to the idea of identifying proteins besides collagen, while the rest of the paragraph (line 30 to line 4) seems out of context and irrelevant here and could be deleted. On the other end, lines 7 to 9 refer to the Non-invasive techniques paragraph, so should be written together.

This paragraph could be developed as well to provide more references about analysis on non-collagenous proteins. Given the proteins identified in this study, work done on ceramics, mummies (for ex: Corthals et al, 2012; Jones et al, 2016) or artwork (numerous studies on paintings with egg and milk binders, work on animal membranes Popowich et al, 2018) would be more appropriate references.

Please provide references on the development of your eZooMS technique and your own work on parchments to give more context to the approach taken here.

"The technique has now been expanded": did that require any technical improvement?

Results:

P 8, L 27: Did the LC-MS/MS confirm that identification and did you identify collagen from another species in LC-MS/MS that could come from animal glue?

Figure 3: Indicate markers that allow identification of sheep, and/or a sheep reference spectrum
SM Table 1: Table difficult to read, with no logical order. Are the proteins classified in terms of frequency, score?

LC-MS/MS results: P9, Line 8 "Differential expression therefore was not taken into account in this study (merely presence/absence)". The manuscript would benefit from such analysis. As far as non-human proteins are concerned, peptide tables should be provided with details of the peptide identification for each category of proteins and some MS/MS spectra (for ex for honey). For human proteins, a general protein table would help to show where these proteins were identified in relation to the sampling.

Egg peptides should be added in Figure 5 too.

P11, L14: "miscarraige" instead of miscarriage

P11: Milk. You say milk was found in WT07 but figure 5 indicates casein in multiple samples.

P11, L41: what is the difference between Legumin B and Legumin type B? Also re-order figures in Fig 5 to logically follow the order referenced in text

P12: cereals. In which samples?

Figure 5: in legend, indicate full names of proteins and species

P14: Human proteins. Which samples? Were the proteins more abundant in stained samples, compared to non-stained? Are there proteins that are specific to CVF?

Table 1: "48" from the Venn diagram, I count 47; "54" from the Venn diagram, I count 55

Furthermore Table 1 is redundant with the venn diagram and can be eliminated. The first two columns are already mentioned in the text

Control sample: explain your choice of an 18th c parchment for control. It would have been more appropriate to use a modern reference of sheep parchment without its own use history.

Conclusion:

L24: "Surface sampling preferentially extracts substances that have been deposited on the document and avoids the inherent bias from predominant proteins when using physical destructive samples". Do you have data comparing the different sampling methods to support this? Surface sampling has the advantage of being less destructive and in such cases the only acceptable sampling method, but you could be missing proteins that have been absorbed in the parchment, or, as in the case of this object, evidence of molecules from earlier uses. Filtration and separation techniques with LC-MS/MS are efficient enough to be able to separate the predominant proteins from minor ones.

Reviewer: 2

Comments to the Author(s)

Please see attached file.

===PREPARING YOUR MANUSCRIPT===

===PREPARING YOUR REVISION IN SCHOLARONE===

Author's Response to Decision Letter for (RSOS-202055.R0)

See Appendix B.

RSOS-202055.R1 (Revision)

Review form: Reviewer 1

Is the manuscript scientifically sound in its present form?

Yes

Are the interpretations and conclusions justified by the results?

Yes

Is the language acceptable?

Yes

Do you have any ethical concerns with this paper?

No

Have you any concerns about statistical analyses in this paper?

No

Recommendation?

Accept as is

Comments to the Author(s)

All appropriate corrections have been made. I have no further comments

Decision letter (RSOS-202055.R1)

Dear Dr Fiddymment,

It is a pleasure to accept your manuscript entitled "Girding the loins? Direct evidence of the use of a medieval English parchment birthing girdle from biomolecular analysis" in its current form for publication in Royal Society Open Science. The comments of the reviewer(s) who reviewed your manuscript are included at the foot of this letter.

You can expect to receive a proof of your article in the near future. Please contact the editorial office (openscience@royalsociety.org) and the production office (openscience_proofs@royalsociety.org) to let us know if you are likely to be away from e-mail

contact – if you are going to be away, please nominate a co-author (if available) to manage the proofing process, and ensure they are copied into your email to the journal.

on behalf of Prof Malcolm White (Subject Editor)
openscience@royalsociety.org

Reviewer comments to Author:
Reviewer: 1

Comments to the Author(s)
All appropriate corrections have been made. I have no further comments

Appendix A

Comments on MS RSOS-202055, “Girding the loins? Direct evidence of the use of a medieval parchment birthing girdle from biomolecular analysis”

General comments

This is an extraordinarily original study, drawing on an impressive survey of the latest work in the history of medieval women’s medicine and the new techniques of scientific study of manuscript books and the material substances used in their manufacture or deployment. Advances in palaeoproteomics have been put to good effect, and the use of a dry, non-invasive technique to extract samples is both sustainable and ethically laudable. Moreover, the choice of this particular object—associated with an event that everyone in the world has experienced once (our own births) and some of use multiple times—is an excellent example to showcase the kinds of ways the palaeosciences are producing knowledge of broad interest. And the fact that the authors have gone to such pains to ground this science in a well-researched cultural framework is most laudable.

I recommend the study for publication. In terms of the science, my biggest remaining question (articulated below) is why the authors did not test for human proteins of other fluids besides cervico-vaginal fluid. I have additional questions about the cultural contextualization. These are intended mostly as queries the authors may wish to consider. I recognize that the main point of the study is to present the physical analysis of the scroll, and not solve every question about 15th/16th century birthing practices. But I do think there are some points that may help flesh out the significance of the analysis.

In what follows, I suggest some ways in which a wider range of literature available might help further contextualize not only this study, but the ways in which scientific approaches might better converse with the interests of historians (including historians of art and religion) who primarily focus on the cultural content of such objects. Not all these studies need necessarily be cited. But it would be good to see recognition by the authors that the most readily discoverable literature in bibliographic databases often does not reflect the most pertinent research that’s been done.

First is the issue of defining a chronological “window” in which the protein evidence might have become embedded in the object. One of the things palaeoproteomic techniques cannot yet do is assess the timeframe of reuses of objects. Whereas the manufacture of the roll itself can be dated paleographically, there is no means to circumscribe a chronological range for the proteins. This is significant, because it’s a way in which protein studies differ from aDNA, for which molecular clock dating mechanisms (imperfect though they still are) have been developed.

In fact, the authors already give clues about the chronological window in which the object was used, and it would be good to make that evidence and its implications more explicit. The authors refer in passing (p. 4, lines 12-14) to “Pre-Reformation English devotion,” which suggests that they are limiting the historical context of use of the scroll to the pre-1536 era. Presumably, after ca. 1536, the scroll was put away somewhere and never used again. But can we be sure? Some further information on what is known of provenance history might help. Still, it seems plausible that if it was used, it would have been within a fairly narrow window of a few decades. Better

explaining what we know now about assessing habits of use of manuscripts might strengthen this portion of the analysis. On the question of physical interactions with devotional objects, art-historian Katy Rudy has been pioneering a science-based approach for a number of years. Yet her work is never cited here. She has more recent work available, but I would recommend her study from 2011 for its conceptual framing of the issues. (All citations can be found at the end of these comments.)

I also think a bit more might be said about maternal mortality in the period, because it is crucial context for the religious/emotional attitudes engendered by childbirth in the period. The cited study by Podd is indeed revealing for reproductive survival of the most elite women in England, but it's hard to believe that those low rates were replicated at other class levels. I note Fleck-Deerderian *et al.*'s recent study on maternal mortality from plague, simply to stress that, in the late 15th and 16th centuries, women in England would still have been affected by heightened infectious disease burden.

Another issue of properly contextualizing both the document and the materials substances found on it has to do with the choice of the "comparison text." The authors have used the Latin *Trotula* text as the main "authority" for childbirth practices. That is a reasonable choice in one sense, since Green established that it was, in fact, the most widely disseminated text on women's medicine in medieval Europe. However, not only did Green establish that major portions of the text were being recycled from much older written authorities (i.e., only certain parts of the *Trotula*—which is in fact an *ensemble* and not the work of a single author or context—could be considered "fresh" empirical data when the texts were written in the 12th century, with an even smaller portion coming from experienced female birth attendants), but Green also established that the Latin *Trotula*'s predominance was waning by the 15th century (when the present scroll was created).

In many areas, newer vernacular adaptations of the *Trotula*, or other ob/gyn works, were becoming more popular, several of which clearly incorporated current medical practices. And most of these had quite a bit more obstetrical content than the *Trotula* had had. Hence, I would have thought that the Middle English *Sickness of Women* Version 2 (edited by Green and Mooney in 2006) would have served as a more appropriate reference. (It is cited several times here, but only sporadically.) There are, for example, more than 30 references in *SoW2* to uses of "hony." The printed volume in which the edition appears includes a comprehensive glossary, and a PDF of the text is posted online making it readily word-searchable. I would not insist on this revision as a point of acceptance, but given the wealth of ob/gyn material available in Middle English (Green 1992; Green 2000), it seems a pity to pass up this opportunity to fully contextualize the data *by comparison with contemporary records* rather than the much earlier 12th-century composite *Trotula* or the much later 17th-century sources that are also used (refs. 38, 40-42).

Finally, there is the question of whether this girdle was used *exclusively* for childbirth. As the authors note, most scrolls are "generic": they are recommended for averting harm in battle and other potentially dangerous scenarios. Additionally (and again a point that the authors acknowledge), there is evidence that these scrolls (and other birth aids) were kept at monasteries and churches, and were loaned out for confinements. In other words, the items passed most of

their existence (we must assume) outside of birth contexts—and far beyond the hands of women. Note, for example, this inherent gender contrast on p. 4, “**Midwives** deployed parchment amulets, precious stones and plant-based remedies during childbirth; the list of **items that the church lent out** to pregnant women is extensive.” Those monks and clerics presumably ate cheese and beans, etc. Can it be ruled out that the substances found on the girdle simply don’t reflect sloppy transportation and storage of the item?

I think the latter scenario less likely, but more because of the persuasiveness of scholarship like Rudy’s (mentioned above) than the evidence presented here. The authors do not cite Green’s 2008 essay, “Gendering the History of Women’s Healthcare,” which had several arguments that put it at odds with traditional notions that women “controlled” their healthcare in premodern Europe. Since the present study is now arguing that there is suggestive evidence to confirm use of the scroll in childbirth settings, is there any way in which these findings confirm or conflict with Green’s? After all, Green’s essay was translated into Chinese as a representative sample of top work in the history of Anglophone science (<https://www.mprl-series.mpg.de/studies/11/1/index.html>), and the present study has the potential to open new possibilities for the still difficult task of reconstructing the history of women’s birth experiences.

Minor comments:

- There is no indication in the study’s title of the geography for the creation or use of this object. The abstract specifies that it was made in England. As noted above, it likely had a fairly short window of use in the context of pre-Reformation England. So its specific geographical context matters.
- Throughout: “Trotula” is a book title and should always be italicized.
- P. 4, line 5, “from a more deprived area”: it is not clear what “deprived” means in this context, nor what the comparandum is (i.e., “more deprived” in comparison to which other area?).
- P. 4, line 10, “at what we would today consider an early age”: average age of death is an *average*. Implying that the “average” woman died around age 33, amid data that seems to include infant mortality (when maternal death in childbirth is impossible), is misleading. (And contrary to the findings of Podd 2020, whom they cite elsewhere.) The authors should be encouraged to find a better way to express this.
- P. 4, line 11, “uterine prolapse”: uterine prolapse, generically, is not a fatal condition. It can potentially lead to sepsis or obstructed bowels, which can be fatal. But it is misleading to list it as a condition contributing to a high death toll. Better to identify truly lethal conditions, such as retained placenta or eclampsia.
- P. 4, lines 14-16: re: the litany of saints invoked for childbirth, one might also note St Susanna. The example published by Green 2003 (not cited by the authors) of masses to be said is an example of the fact that the level of concern about childbirth extended well beyond the hour of birth and the birthing room. The emotional/psychological atmosphere of fear is an important context here. Note that Green found the prayers for St Susanna in a MS with an obstetrical text! Note, too, that Green finds another charm for Susanna in Wellcome MS 410, surely worth mentioning with respect to the present investigation.
- P. 5, lines 3-6, “Women’s health, encompassing not just pregnancy and birth but also menstruation and the various health issues shared with men (digestive complaints,

problems with the eyes and ears, pestilential illness, etc.) was often addressed in the arena of oral culture, where religious and magical rituals were prominent”: This statement is problematic in so far as it ignores the ample existence in England of *written* texts about women’s health. That does not diminish the likelihood that there was also, simultaneously, an oral culture difficult to recapture.

- P. 5, lines 27-28, “Perhaps because of the fervor with which the birthing talismans were destroyed by Reformers”: note that this relates to the point made above about the chronological window during which the girdle might have been most actively used. On changes in obstetric practices due to the Protestant Reformation, the authors may wish to consult Fissell 2004. On the longevity of use of the eagle stone which (unlike religious talismans) survived in use past the Reformation, see Phelps Walsh 2014.
- P. 6, lines 14-16, re: “And yf a woman travell wyth chylde gyrdes thys mesure abowte hyr wombe and she shall be delyvyrs wythowte pallelle and the chylde shall have crystendome and the mother puryfycatyon”: the same statement is found in BL Harley Roll T 11, as noted by Green 2003, who was citing Buhler 1964. In the latter case, however, it is followed by an assertion that it can be used with equal success by men going into battle. I stress this simply to reaffirm that it really is important, for the argument of the present paper, that the authors provide *convincing* proof that the biological traces on this object are birth products and not simply human fluids.
- Pp. 7-8, “This holds great promise for || phylogenetic studies where genetic analysis is not possible due to poor preservation of DNA”: The two studies cited in support of this statement (ref. 27: Welker F et al. 2020 The dental proteome of Homo antecessor. Nature; ref. 30: Chen F et al. 2019 A late Middle Pleistocene Denisovan mandible from the Tibetan Plateau. Nature 569) are, as I read them, claiming to *supplement* phylogenetic models derived from genomic studies. The proteins themselves do not contribute to information on the whole genome, and therefore cannot be used to build phylogenetic trees themselves. Perhaps I’m missing something here, but I’m not sure this claim is needed to support the novelty of the present study.
- P. 12, re: legumes: I can’t recall anything in the literature about birth scrolls about when they were to be removed after birth. Immediately after the child comes out? After the placenta emerges? Or does it stay on the woman for several days thereafter? I mention this simply because the items identified can function as medicines, but they are also foods. Restorative foods would be part of the post-partum care.
- P. 14, human proteins: This, of course, is where the analysis really hits paydirt. However, I didn’t understand why there was no analysis of other types of human fluids (blood, oral mucous, sweat). My question (and I know nothing specific about cervical fluids) is: to what extent are those proteins unique to cervico-vaginal fluid, and to what extent are they shared with fluids coming from other parts of the body? Again, these scrolls themselves “advertise” their uses for other life-threatening circumstances, so it is by no means inappropriate to ask for confirmation, not simply that these are human proteins, but that they’re a particular kind of human protein. And why no blood? Perhaps the scrolls were considered precious enough that they would be removed at the first sign of the waters breaking. But if they’re close enough to the vagina to be getting wet with cervico-vaginal fluid, then we would expect blood, too.
- P. 16, lines 33-34: The authors need to clarify to what university the Institute of Medieval and Early Modern Studies is connected.

- Ref. 2: the author's name is given as "In press."

Additional Bibliography:

- Green, Monica H. "'Obstetrical and Gynecological Texts in Middle English,'" *Studies in the Age of Chaucer* 14 (1992), 53-88.
- Green, Monica H. "Medieval Gynecological Texts: A Handlist," in Monica H. Green, *Women's Healthcare in the Medieval West: Texts and Contexts* (Aldershot: Ashgate, 2000), Appendix, pp. 1-36.
- Green, Monica H. "Masses in Remembrance of 'Seynt Susanne': A Fifteenth-Century Spiritual Regimen," *Notes & Queries* n. s. 50, no. 4 (December 2003), 380-84.
- Green, Monica H. "Gendering the History of Women's Healthcare," *Gender and History*, Twentieth Anniversary Special Issue, 20, no. 3 (November 2008), 487-518.
- Fissell, Mary E. "The Politics of Reproduction in the English Reformation," *Representations* 87, no. 1 (2004), 43-81.
- Fleck-Derderian, Shannon, Christina A. Nelson, Katharine M. Cooley, Zachary Russell, Shana Godfred-Cato, Nadia L. Oussayef, Titilope Oduyebo, Sonja A. Rasmussen, Denise J. Jamieson, and Dana Meaney-Delman. "Plague During Pregnancy: A Systematic Review," *Clinical Infectious Diseases* 70, Supplement 1 (21 May 2020), S30-S36.
- Phelps Walsh, Katharine. "Marketing Midwives in Seventeenth-Century London: A Re-examination of Jane Sharp's *The Midwives Book*," *Gender & History* 26, no.2 (August 2014), pp. 223-241.
- Rudy, Kathryn M. "Kissing Images, Unfurling Rolls, Measuring Wounds, Sewing Badges and Carrying Talismans: Considering Some Harley Manuscripts through the Physical Rituals they Reveal," *The Electronic British Library Journal*, 2011, <http://www.bl.uk/eblj/2011/articles/article5.html>.

Appendix B

We would like to thank the reviewers for their constructive comments on our manuscript. The suggestions and critiques have been welcomed and addressed and we believe the paper has been strengthened. Please find our detailed responses below highlighted in red.

Reviewer: 1

Comments to the Author(s)

- More details of the analysis and results could be given in Supplementary Materials, such as detailed tables of peptide and protein identification.
- There is no comparison of the stained and non-stained spots, so it is not clear whether the human proteins were found in these locations only.

Detailed comments:

Abstract:

- It needs re-writing, as it insists on the potential of the method more than on the originality of the case study which is the real focus of the paper. “Dry non-invasive sampling method” is mentioned three times in the abstract, yet this paper is not about the description of a new method. In that regards, the first sentence is misleading. In fact the methods are not described at all, even in Materials and Methods, and instead refer to two published papers. Either the innovative aspect of the extraction, if any, needs to be detailed, or the abstract should clarify that the focus is here rather on the application of already developed methods, and highlights the importance of the results instead.
- Furthermore, the study is about one manuscript and does not represent a large scale study of “stained manuscripts”. While the method could potentially work on all blood-stained parchments, this is an extrapolation that doesn't take into account the possible special preservation circumstances of the studied manuscript (such as heavy and/or repetitive use), which is a point that was not really addressed in the manuscript.
- “both human and non-human peptides from the stains including evidence for the use of honey, cereals, ovicaprine milk and legumes”: in the paper you mention that both stained and non-stained samples were taken, but did not give further detail on which samples proteins were found. Does that mean all these proteins were found in stained areas?

The abstract has been rewritten as follows: “In this paper we describe palaeoproteomic evidence obtained from a stained medieval birth girdle using a previously developed dry non-invasive sampling technique. The parchment birth girdle studied (Wellcome Collection Western MS. 632) was made in England and thought to be used by pregnant women while giving birth. We were able to extract both human and non-human peptides from the manuscript, including evidence for the use of honey, cereals, ovicaprine milk and legumes. In addition, a large number of human peptides were detected on the birth roll, many of which are found in cervico-vaginal fluid. This suggests that the birth roll was actively used during childbirth. This study is the first to extract and analyse non-collagenous peptides from a birth girdle using this sampling method and demonstrates the potential of this type of analysis for stained manuscripts, providing direct biomolecular evidence for active use.”

Keywords: birth and stains are very generic. Perhaps gives more precision: birth girdle, blood stains...

We have changed birth to birth girdle, however in the case of stains we would like to leave it quite broad as we are not referring just to blood stains, but other bodily fluids, food substances, etc

Introduction:

- starting the paragraph with “even today” sounds odd. It feels like the sentence should be reversed. The references on modern birth are of little use here unless you want to make a direct comparison.

Paragraph changed to read: “Childbearing can be a highly perilous time for both mother and child even today (WHO et al. 2019; “Neonatal Mortality - UNICEF DATA” n.d.; Hug et al. 2019), but in medieval Europe...”

The references have been included as we didn’t feel we make such a statement without backing up with evidence, although we agree that we are not using this data to make a direct comparison.

- Line 6 to 8: Even repeated twice and makes the sentence heavy

The first ‘even’ has been replaced with although

- Line 26: “siezed” instead of seized

Changed

- Page 7: the Samples paragraph, as well as the first two sentences of eZooMS from Materials and Methods should be moved at the end of introduction as they contain information to understand the results and a valuable description of the object.

Changed

- Furthermore a sort of map showing where the samples were taken in relation to the stains should be added.

An image of the birth girdle indicating where the samples have been taken has been added to the supplementary material (Supplementary Figure1 and Figure2).

- Page 7 and 8: the two paragraphs about proteomics and non-invasive techniques should be merged into one. Lines 27 to 30 should be linked to lines 9 to 12 since you come back to the idea of identifying proteins besides collagen, while the rest of the paragraph (line 30 to line 4) seems out of context and irrelevant here and could be deleted. On the other end, lines 7 to 9 refer to the Non-invasive techniques paragraph, so should be written together.

The two paragraphs have been merged into one and now reads as follows:

“Recent advances in palaeoproteomics and non-invasive techniques

Recent advances in palaeoproteomic techniques have allowed a much more in-depth examination of biomolecular evidence on different substrates. Proteomic analysis of bone [24–26], tooth enamel [27,28] and shell [29] have proved the survival of a diverse collection of proteins in addition to the primary structural protein. Proteomic analysis has also been successfully carried out on more recent objects but with equally interesting outcomes. One of the obstacles to overcome is an inherent bias to the predominant protein present (in the case of parchment that protein is collagen) and consequently the signal of lower concentration proteins is drowned out, in most cases to such a degree that they are undetectable.

In addition, a further obstacle to access has been the method of sampling. Initial proteomic studies of cultural heritage objects has always required taking physical destructive samples [31], severely limiting the amount of objects that can be subjected to this kind of analysis. Recently there has been a move to less invasive forms of sampling. The use of non-invasive EVA diskettes...”

- This paragraph could be developed as well to provide more references about analysis on non-collagenous proteins. Given the proteins identified in this study, work done on ceramics, mummies (for ex: Corthals et al, 2012; Jones et al, 2016) or artwork (numerous studies on paintings with egg and milk binders, work on animal membranes Popowich et al, 2018) would be more appropriate references.

The paragraph has been amended to include more relevant citations as follows:

“...Proteomic analysis of bone [24–26], tooth enamel [27,28], shell [29] and mummified skin tissue [30] have proved the survival of a diverse collection of proteins in addition to the primary structural protein. Proteomic analysis has also been successfully carried out on more recent objects like artwork [31,32], textiles [33] and manuscripts [34,35] with equally interesting outcomes. However, one of the obstacles...”

- Please provide references on the development of your eZooMS technique and your own work on parchments to give more context to the approach taken here.

Citation added. Paragraph now reads:

“Here we report the first proteomic analysis of a historic parchment document using a dry non-invasive sampling technique (eZooMS). This technique was developed for use on parchment documents allowing for the extraction not only of collagen (Fiddyment et al. 2015) but also DNA both from the animal and the microbiome (Teasdale et al. 2017). This sampling technique has now been used to analyse the broader set of proteins present on the surface of the document which can provide information about the history and use of this object.”

- “The technique has now been expanded”: did that require any technical improvement?

We have changed the word *expanded* to *used*.

Results:

- P 8, L 27: Did the LC-MS/MS confirm that identification and did you identify collagen from another species in LC-MS/MS that could come from animal glue?
Sheep collagen was detected in the LCMSMS data, however, this is a general problem we have with collagen identification through LCMSMS. As collagen sequences are highly conserved, results often appear for closely related species (including some extinct species!). In addition, some species are better characterized in the database than others, *Bos taurus* being an obvious candidate, which is often identified when sheep is run as it is a better curated sequence.
For collagen analysis PMF provides a cleaner signal (that also allows for the detection of glues etc) and in this case all the samples were identified as sheep.
- Figure 3: Indicate markers that allow identification of sheep, and/or a sheep reference spectrum
Figure 3 has been modified to reflect the markers used to identify the parchment as sheep
- SM Table 1: Table difficult to read, with no logical order. Are the proteins classified in terms of frequency, score?

Proteins are now listed in alphabetical order of protein name.

- LC-MS/MS results: P9, Line 8 “Differential expression therefore was not taken into account in this study (merely presence/absence)”. The manuscript would benefit from such analysis. As far as non-human proteins are concerned, peptide tables should be provided with details of the peptide identification for each category of proteins and some MS/MS spectra (for ex for honey). For human proteins, a general protein table would help to show where these proteins were identified in relation to the sampling. While we agree that a differential expression analysis would be interesting it is out of the scope of this current study but we hope to include this type of analysis in future follow up studies.

We have now added an additional supplementary file listing all the peptide identifications.

We have also added a table of the human proteins showing in which samples they have been detected.

We have had problems exporting MS/MS spectra and at the moment are unable to provide them, however all the raw data and Maxquant results files will be made available and can therefore be visualised in the Maxquant software.

- Egg peptides should be added in Figure 5 too.
We are quite limited in space in Figure 5 as including any other plots would make the others smaller and unreadable. We decided not to include egg as some of the peptides were also found in the control so we could not say it was not contamination, and the presence of egg white could be due to parchment processing techniques.

- P11, L14: “miscarraige” instead of miscarriage

Changed

- P11: Milk. You say milk was found in WT07 but figure 5 indicates casein in multiple samples.

WT07 presented the largest amount of milk peptides, but I have amended the text as follows: “Ovicaprine milk proteins were predominantly detected in WT07 (although certain peptides were also present in other samples) including: alpha-casein 1 (five specific peptides), alpha-casein 2 (two peptides) and beta-casein (five peptides).”

- P11, L41: what is the difference between Legumin B and Legumin type B? Also re-order figures in Fig 5 to logically follow the order referenced in text

Legumin is the name given to the protein in *Pisum sativum* (Garden pea) whereas Legumin type B is the name given to the equivalent in *Vicia faba* (Broad bean) (*Faba vulgaris*). The text has been amended as follows: “In addition we also detected the presence of peptides from two other proteins, Legumin A (two peptides) and Legumin B/type B (ten peptides) although the specificity to organism can not be differentiated between broad bean, garden peas and common vetch.”

- P12: cereals. In which samples?

Peptides for cereals have been detected in all birth girdle samples, but especially in sample WT02. The text now reads: “We were also able to detect the presence of cereals in all the birth girdle samples (although mostly in sample WT02).”

- Figure 5: in legend, indicate full names of proteins and species

Figure 5 has been changed to include the full names of the proteins (and in the same order as the text) however the species has not been included as for some proteins it has not been possible to discriminate between species (e.g. ovicaprine milk, legumes, etc)

- P14: Human proteins. Which samples? Were the proteins more abundant in stained samples, compared to non-stained? Are there proteins that are specific to CVF?
We have now included supplementary table 2 which list the human proteins and in which samples they were detected. We detected more human proteins in the stained samples than the unstained, however we only had one unstained sample and seven stained samples. There doesn't seem to be proteins specific to CVF but when we compared our samples to other proteomes (skin, saliva, amniotic fluid and blood) only blood matched some of the proteins, but to a lesser extent than CVF so we concluded that the most likely explanation is that the proteins come from CVF.
- Table 1: "48" from the Venn diagram, I count 47; "54" from the Venn diagram, I count 55. Furthermore Table 1 is redundant with the venn diagram and can be eliminated. The first two columns are already mentioned in the text.
Table has been removed and numbers amended.
- Control sample: explain your choice of an 18th c parchment for control. It would have been more appropriate to use a modern reference of sheep parchment without its own use history.
The choice of this parchment is partly due to its accessibility. This is a parchment document that has been donated to us for use in destructive analysis and has been used as a control or comparison parchment for many of our experiments. We prefer to use this parchment rather than a modern one as it will more likely have been processed using traditional methods and in addition it will have its own history of use which we felt was more comparable than a modern pristine piece of parchment. It means we may have been overly cautious when eliminating proteins from our comparison but we felt it better reflected the possible environmental or historical proteins that may be deposited on parchment merely by storage and age.

Conclusion:

- L24: "Surface sampling preferentially extracts substances that have been deposited on the document and avoids the inherent bias from predominant proteins when using physical destructive samples". Do you have data comparing the different sampling methods to support this? Surface sampling has the advantage of being less destructive and in such cases the only acceptable sampling method, but you could be missing proteins that have been absorbed in the parchment, or, as in the case of this object, evidence of molecules from earlier uses. Filtration and separation techniques with LC-MS/MS are efficient enough to be able to separate the predominant proteins from minor ones.
This is something that we want to explore in more depth, however in our previous experiments we have found that using destructive samples gives an overwhelming predominance of collagen as it is the majority protein. Although filtration and separation are possible, they are more time consuming and costly than simply using an eraser for sampling and we have seen a difference in results when using this method. Furthermore this method can be performed by conservators, as here, using tools and materials available in all conservation studios. However, we would like to explore the possibility of looking for these minor proteins in destructive samples also, although these are usually just experimental samples as destructive sampling is hardly ever allowed for historic documents.

Reviewer: 2

General comments

This is an extraordinarily original study, drawing on an impressive survey of the latest work in the history of medieval women's medicine and the new techniques of scientific study of manuscript books and the material substances used in their manufacture or deployment. Advances in palaeoproteomics have been put to good effect, and the use of a dry, non-invasive technique to extract samples is both sustainable and ethically laudable. Moreover, the choice of this particular object—associated with an event that everyone in the world has experienced once (our own births) and some of use multiple times—is an excellent example to showcase the kinds of ways the palaeosciences are producing knowledge of broad interest. And the fact that the authors have gone to such pains to ground this science in a well-researched cultural framework is most laudable.

I recommend the study for publication. In terms of the science, my biggest remaining question (articulated below) is why the authors did not test for human proteins of other fluids besides cervico-vaginal fluid. I have additional questions about the cultural contextualization. These are intended mostly as queries the authors may wish to consider. I recognize that the main point of the study is to present the physical analysis of the scroll, and not solve every question about 15th/16th century birthing practices. But I do think there are some points that may help flesh out the significance of the analysis.

In what follows, I suggest some ways in which a wider range of literature available might help further contextualize not only this study, but the ways in which scientific approaches might better converse with the interests of historians (including historians of art and religion) who primarily focus on the cultural content of such objects. Not all these studies need necessarily be cited. But it would be good to see recognition by the authors that the most readily discoverable literature in bibliographic databases often does not reflect the most pertinent research that's been done.

On only examining literature in 'bibliographic databases': in the original article, much more information was cited, but needed to be cut down for length. The suggested reading was enormously helpful. We have made an effort to incorporate the material suggested by the Reviewer.

First is the issue of defining a chronological "window" in which the protein evidence might have become embedded in the object. One of the things palaeoproteomic techniques cannot yet do is assess the timeframe of reuses of objects. Whereas the manufacture of the roll itself can be dated paleographically, there is no means to circumscribe a chronological range for the proteins. This is significant, because it's a way in which protein studies differ from aDNA, for which molecular clock dating mechanisms (imperfect though they still are) have been developed.

In fact, the authors already give clues about the chronological window in which the object was used, and it would be good to make that evidence and its implications more explicit. The authors refer in passing (p. 4, lines 12-14) to "Pre-Reformation English devotion," which suggests that they are limiting the historical context of use of the scroll to the pre-1536 era. Presumably, after ca. 1536, the scroll was put away somewhere and never used again. But can we be sure? Some further

information on what is known of provenance history might help. Still, it seems plausible that if it was used, it would have been within a fairly narrow window of a few decades. Better explaining what we know now about assessing habits of use of manuscripts might strengthen this portion of the analysis. On the question of physical interactions with devotional objects, art historian Katy Rudy has been pioneering a science-based approach for a number of years. Yet her work is never cited here. She has more recent work available, but I would recommend her study from 2011 for its conceptual framing of the issues. (All citations can be found at the end of these comments.)

Rudy was invaluable, as was Fissell, and their sources, for determining the chronological window of use of MS 632, especially for 'assessing habits of use of manuscripts'. We discovered that the use of birth girdles were specifically targeted in post-Reformation Britain [Fissell], but with good cause as these were known to be used on the sly [Thomas]. Rudy's work was extremely helpful in framing the excessive use of MS 632 within larger material culture. The authors would like to thank the Reviewer for pointing out these works. The production of MS 632 cannot be dated more specifically than the latter decades of the 15th century or early years of the 16th century, and no evidence of early provenance is extant. The Reformation does provide a logical cut-off point for usage, but clandestine continuing use after 1536 (even for some time) is possible. We have added a short section to address the chronological window of use.

I also think a bit more might be said about maternal mortality in the period, because it is crucial context for the religious/emotional attitudes engendered by childbirth in the period. The cited study by Podd is indeed revealing for reproductive survival of the most elite women in England, but it's hard to believe that those low rates were replicated at other class levels. I note Fleck Derderian *et al.*'s recent study on maternal mortality from plague, simply to stress that, in the late 15th and 16th centuries, women in England would still have been affected by heightened infectious disease burden.

This paragraph has been rewritten to take into account the atmosphere and included additional sources on mortality, as well as the Reviewer's excellent point that pregnant women have heightened infectious disease burden.

Another issue of properly contextualizing both the document and the materials substances found on it has to do with the choice of the "comparison text." The authors have used the Latin *Trotula* text as the main "authority" for childbirth practices. That is a reasonable choice in one sense, since Green established that it was, in fact, the most widely disseminated text on women's medicine in medieval Europe. However, not only did Green establish that major portions of the text were being recycled from much older written authorities (i.e., only certain parts of the *Trotula*—which is in fact an *ensemble* and not the work of a single author or context—could be considered "fresh" empirical data when the texts were written in the 12th century, with an even smaller portion coming from experienced female birth attendants), but Green also established that the Latin *Trotula*'s predominance was waning by the 15th century (when the present scroll was created).

In many areas, newer vernacular adaptations of the *Trotula*, or other ob/gyn works, were becoming more popular, several of which clearly incorporated current medical

practices. And most of these had quite a bit more obstetrical content than the *Trotula* had had. Hence, I would have thought that the Middle English *Sickness of Women* Version 2 (edited by Green and Mooney in 2006) would have served as a more appropriate reference. (It is cited several times here, but only sporadically.) There are, for example, more than 30 references in *SoW2* to uses of “hony.” The printed volume in which the edition appears includes a comprehensive glossary, and a PDF of the text is posted online making it readily word-searchable. I would not insist on this revision as a point of acceptance, but given the wealth of ob/gyn material available in Middle English (Green 1992; Green 2000), it seems a pity to pass up this opportunity to fully contextualize the data *by comparison with contemporary records* rather than the much earlier 12th-century composite *Trotula* or the much later 17th-century sources that are also used (refs. 38, 40-42).

This seems a sensible suggestion and *Sickness* has been incorporated as initial text.

Finally, there is the question of whether this girdle was used *exclusively* for childbirth. As the authors note, most scrolls are “generic”: they are recommended for averting harm in battle and other potentially dangerous scenarios. Additionally (and again a point that the authors acknowledge), there is evidence that these scrolls (and other birth aids) were kept at monasteries and churches, and were loaned out for confinements. In other words, the items passed most of their existence (we must assume) outside of birth contexts—and far beyond the hands of women. Note, for example, this inherent gender contrast on p. 4, “**Midwives** deployed parchment amulets, precious stones and plant-based remedies during childbirth; the list of **items that the church lent out** to pregnant women is extensive.” Those monks and clerics presumably ate cheese and beans, etc. Can it be ruled out that the substances found on the girdle simply don’t reflect sloppy transportation and storage of the item?

We cannot say for certain that the proteins detected were not environmental, however given the use of the birth girdle and the high level of human proteins detected (which best matched CVF) it would indicate that the girdle was actively used. If the girdles were lent out to various users, it is more likely that the proteins detected come from the activities of the user rather than where it was stored. As birthing girdles were considered sacred items of veneration, it seems unlikely for them to have been stored in the refectory. A more likely assumption would be that it was stored somewhere with veneration, to keep it ‘clean’ and intact for the next user. However it is true that we cannot distinguish between these cases and therefore cannot guarantee that the plant proteins are not environmental contamination.

I think the latter scenario is less likely, but more because of the persuasiveness of scholarship like Rudy’s (mentioned above) than the evidence presented here. The authors do not cite Green’s 2008 essay, “Gendering the History of Women’s Healthcare,” which had several arguments that put it at odds with traditional notions that women “controlled” their healthcare in premodern Europe. Since the present study is now arguing that there is suggestive evidence to confirm use of the scroll in childbirth settings, is there any way in which these findings confirm or conflict with Green’s? After all, Green’s essay was translated into Chinese as a representative sample of top work in the history of Anglophone science (<https://www.mprl-series.mpg.de/studies/11/1/index.html>), and the present study has the potential to open new possibilities for the still difficult task of reconstructing

the history of women's birth experiences.

Duffy's evidence shows that much of what was seized during the Reformation had to do with helping women through childbirth (not all girdles). The argument could be made that, as most monasteries were run by men, their lending (or 'renting out') of the birth girdles was male-control. But the lists in which we find most mention of birth-scrolls is in the Dissolution records, not in specific use of birthing-girdles, making more precise judgements difficult to determine. Post-Reformation England, and the sudden male clerical control over what items women were /were not allowed to use during childbirth, may speak to Green's argument better, but this seems outside the remit of the essay.

Minor comments:

- There is no indication in the study's title of the geography for the creation or use of this object. The abstract specifies that it was made in England. As noted above, it likely had a fairly short window of use in the context of pre-Reformation England. So its specific geographical context matters.

Done

- Throughout: "Trotula" is a book title and should always be italicized.
Done
- P. 4, line 5, "from a more deprived area": it is not clear what "deprived" means in this context, nor what the comparandum is (i.e., "more deprived" in comparison to which other area?).
Done
- P. 4, line 10, "at what we would today consider an early age": average age of death is an *average*. Implying that the "average" woman died around age 33, amid data that seems to include infant mortality (when maternal death in childbirth is impossible), is misleading. (And contrary to the findings of Podd 2020, whom they cite elsewhere.) The authors Done
- P. 4, line 11, "uterine prolapse": uterine prolapse, generically, is not a fatal condition. It can potentially lead to sepsis or obstructed bowels, which can be fatal. But it is misleading to list it as a condition contributing to a high death toll. Better to identify truly lethal conditions, such as retained placenta or eclampsia.
Done
- P. 4, lines 14-16: re: the litany of saints invoked for childbirth, one might also note St Susanna. The example published by Green 2003 (not cited by the authors) of masses to be said is an example of the fact that the level of concern about childbirth extended well beyond the hour of birth and the birthing room. The emotional/psychological atmosphere of fear is an important context here. Note that Green found the prayers for St Susanna in a MS with an obstetrical text! Note, too, that Green finds another charm for Susanna in Wellcome MS 410, surely worth mentioning with respect to the present investigation.

References to St Susanna and atmosphere of fear have been incorporated. It

is slightly difficult to incorporate MS 410, as at the mention of the saints invoked, the paper does not yet reveal this is Wellcome MS 632. Will the Reviewer allow me to use the phrase 'beyond the hour of birth in the birthing room' in the paper?

- P. 5, lines 3-6, "Women's health, encompassing not just pregnancy and birth but also menstruation and the various health issues shared with men (digestive complaints, problems with the eyes and ears, pestilential illness, etc.) was often addressed in the arena of oral culture, where religious and magical rituals were prominent": This statement is problematic in so far as it ignores the ample existence in England of *written* texts about women's health. That does not diminish the likelihood that there was also, simultaneously, an oral culture difficult to recapture.

We completely agree with this point - we have modified the text.

- P. 5, lines 27-28, "Perhaps because of the fervor with which the birthing talismans were destroyed by Reformers": note that this relates to the point made above about the chronological window during which the girdle might have been most actively used. On changes in obstetric practices due to the Protestant Reformation, the authors may wish to consult Fissell 2004. On the longevity of use of the eagle stone which (unlike religious talismans) survived in use past the Reformation, see Phelps Walsh 2014.

The changes in birthing practices, identified by Walsh and Fissell, have been incorporated to testify to the chronological window of the use of the birth girdle.

- P. 6, lines 14-16, re: "And yf a woman travell wyth chylde gyrdes thys mesure abowte hyr wombe and she shall be delyvyrs wythowte parelle and the chylde shall have crystendome and the mother puryfycatyon": the same statement is found in BL Harley Roll T 11, as noted by Green 2003, who was citing Buhler 1964. In the latter case, however, it is followed by an assertion that it can be used with equal success by men going into battle. I stress this simply to reaffirm that it really is important, for the argument of the present paper, that the authors provide *convincing* proof that the biological traces on this object are birth products and not simply human fluids.

See the following section.

- Pp. 7-8, "This holds great promise for || phylogenetic studies where genetic analysis is not possible due to poor preservation of DNA": The two studies cited in support of this statement (ref. 27: Welker F et al. 2020 The dental proteome of Homo antecessor. Nature; ref. 30: Chen F et al. 2019 A late Middle Pleistocene Denisovan mandible from the Tibetan Plateau. Nature 569) are, as I read them, claiming to *supplement* phylogenetic models derived from genomic studies. The proteins themselves do not contribute to information on the whole genome, and therefore cannot be used to build phylogenetic trees themselves. Perhaps I'm missing something here, but I'm not sure this claim is needed to support the novelty of the present study.

In the studies cited proteins have been used in cases where DNA is irretrievable due to the age of the remains. Although you cannot analyse the entire genome through proteomics it is possible to track mutations in the protein sequence and therefore perform a phylogenetic analysis of that

protein, albeit more limited than the genetic ones. This can prove very useful in cases where DNA has not been recovered, which is what we tried to highlight, however we have removed the sentence to avoid confusion.

- P. 12, re: legumes: I can't recall anything in the literature about birth scrolls about when they were to be removed after birth. Immediately after the child comes out? After the placenta emerges? Or does it stay on the woman for several days thereafter? I mention this simply because the items identified can function as medicines, but they are also foods. Restorative foods would be part of the post-partum care.

The literature is ambiguous about how to use the birth girdle - we know of no contemporary accounts that describe when it is to be removed. The larger point the reviewer makes is that legumes, or all of these food products, may not necessarily indicate birthing remedies. The authors have amended this more specifically to say that it is interesting to note that all of the ingredients were used in birthing procedures, but their presence on the parchment could be due to other means (e.g. postpartum care).

- P. 14, human proteins: This, of course, is where the analysis really hits paydirt. However, I didn't understand why there was no analysis of other types of human fluids (blood, oral mucous, sweat). My question (and I know nothing specific about cervical fluids) is: to what extent are those proteins unique to cervico-vaginal fluid, and to what extent are they shared with fluids coming from other parts of the body? Again, these scrolls themselves "advertise" their uses for other life-threatening circumstances, so it is by no means inappropriate to ask for confirmation, not simply that these are human proteins, but that they're a particular kind of human protein. And why no blood? Perhaps the scrolls were considered precious enough that they would be removed at the first sign of the waters breaking. But if they're close enough to the vagina to be getting wet with cervico-vaginal fluid, then we would expect blood, too.

When doing the analysis we compared our samples to other proteomes (skin, saliva, amniotic fluid and blood) but none of these matched our sample proteins with the exception of blood. However it matched fewer proteins than CVF, and those it did much were shared with CVF (as they are common physiological proteins) so we concluded that the most likely explanation is that the proteins come from CVF.

- P. 16, lines 33-34: The authors need to clarify to what university the Institute of Medieval and Early Modern Studies is connected.

Done

- Ref. 2: the author's name is given as "In press."

This article is published on the Unicef Website and there is no author cited. We have changed this to 'Anonymous'.

Additional Bibliography:

Green, Monica H. "Obstetrical and Gynecological Texts in Middle English," *Studies in the Age of Chaucer* 14 (1992), 53-88.

Green, Monica H. "Medieval Gynecological Texts: A Handlist," in Monica H. Green,

- Women's Healthcare in the Medieval West: Texts and Contexts* (Aldershot: Ashgate, 2000), Appendix, pp. 1-36.
- Green, Monica H. "Masses in Remembrance of 'Seynt Susanne': A Fifteenth-Century Spiritual Regimen," *Notes & Queries* n. s. 50, no. 4 (December 2003), 380-84.
- Green, Monica H. "Gendering the History of Women's Healthcare," *Gender and History*, Twentieth Anniversary Special Issue, 20, no. 3 (November 2008), 487-518. Fissell, Mary E. "The Politics of Reproduction in the English Reformation," *Representations* 87, no. 1 (2004), 43-81.
- Fleck-Derderian, Shannon, Christina A. Nelson, Katharine M. Cooley, Zachary Russell, Shana Godfred-Cato, Nadia L. Oussayef, Titilope Oduyebo, Sonja A. Rasmussen, Denise J. Jamieson, and Dana Meaney-Delman. "Plague During Pregnancy: A Systematic Review," *Clinical Infectious Diseases* 70, Supplement 1 (21 May 2020), S30-S36.
- Phelps Walsh, Katharine. "Marketing Midwives in Seventeenth-Century London: A Re examination of Jane Sharp's *The Midwives Book*," *Gender & History* 26, no.2 (August 2014), pp. 223-241.
- Rudy, Kathryn M. "Kissing Images, Unfurling Rolls, Measuring Wounds, Sewing Badges and Carrying Talismans: Considering Some Harley Manuscripts through the Physical Rituals they Reveal," *The Electronic British Library Journal*, 2011, <http://www.bl.uk/eblj/2011/articles/article5.html>.